# How Habitat Simplification Shapes the Morphological Characteristics of Ant Assemblages (Hymenoptera: Formicidae) in Different Biogeographical Contexts

**DOI:** 10.3390/insects15120961

**Published:** 2024-12-03

**Authors:** Ana Cristina da Silva Utta, Gianpasquale Chiatante, Enrico Schifani, Alberto Meriggi, Itanna Oliveira Fernandes, Paulo A. V. Borges, Ricardo R. C. Solar, Fabricio Beggiato Baccaro, Donato Antonio Grasso

**Affiliations:** 1Programa de Pós-Graduação em Ecologia, Instituto Nacional de Pesquisas da Amazônia (INPA), Manaus CEP 69067-375, Amazonas, Brazil; 2Department of Ecological and Biological Sciences, University of Tuscia, Largo dell’Università s/n, 01100 Viterbo, Italy; 3Department of Chemistry, Life Sciences and Environmental Sustainability, University of Parma, Parco Area delle Scienze 11/a, 43124 Parma, Italydonato.grasso@unipr.it (D.A.G.); 4Dipartimento di Scienze della Terra e dell’Ambiente, Università di Pavia, via Ferrata 1, 27100 Pavia, Italy; alberto.meriggi@unipv.it; 5Programa de Pós-Graduação em Entomologia, Instituto Nacional de Pesquisas da Amazônia (INPA), Manaus CEP 69060-001, Amazonas, Brazil; 6Azorean Biodiversity Group, cE3c-Centre for Ecology, Evolution and Environmental Changes, CHANGE–Global Change and Sustainability Institute, School of Agricultural and Environmental Sciences, University of the Azores, Rua Capitão João d’Ávila, Pico da Urze, 9700-042 Angra do Heroísmo, Azores, Portugal; paulo.av.borges@uac.pt; 7Departamento de Genética, Ecologia e Evolução, Instituto de Ciências Biológicas, Universidade Federal de Minas Gerais (UFMG), Belo Horizonte CEP 31270-901, Minas Gerais, Brazil; rrsolar@gmail.com; 8Departamento de Biologia, Instituto de Ciências Biológicas, Universidade Federal do Amazonas (UFAM), Manaus CEP 69080-900, Amazonas, Brazil

**Keywords:** ants, biotic homogenization, functional traits, Brazil, Italy

## Abstract

Changes in how land is used by humans can greatly affect ant communities, often causing a decrease in the variety of species or making their composition more similar. By studying ant body structures, scientists can better understand how these changes impact ant diversity. This study looked at ant communities in two environments—agricultural areas and secondary forests—in Italy and the Brazilian Amazon. The research occurred in the Ticino River Natural Park in Italy, in 15 agricultural and 12 forested sample site areas, and in the Paragominas mosaic area in Pará, Brazil, in 15 agricultural sites and 13 forest sites. The study found that secondary forests in both countries have more ant species than agricultural areas. Overall, the diversity of ant traits in Brazil was greater than in Italy, but agricultural areas in both countries showed similar levels of trait diversity. When comparing forests with the same number of species, Brazilian forests showed a wider range of traits than Italian forests. Agricultural landscapes tend to make ant communities more similar. At the same time, secondary forests display greater differences, emphasizing the impact of environmental conditions in shaping these communities depending on how the land is used.

## 1. Introduction

Changes in land cover and land use caused by humans can strongly influence species community structure [1,2], frequently associated with the decline in taxonomic diversity or homogenization of species composition [3]. This phenomenon usually involves the replacement of unique species with more widespread species, leading to a loss of biodiversity and a shift towards a more uniform biotic composition [4,5,6,7,8].

Biotic homogenization can generally encompass functional homogenization, where the taxonomic and functional species diversity decrease, leading to more uniform and less diverse community structures [9,10]. Functional traits can be morphological, biochemical, physiological, structural, phenological, or behavioral features of organisms linked with ecosystem functions and processes, ultimately impacting their fitness [11]. They offer a complementary tool that allows investigating assemblages with no shared species [12,13] and link species performance to ecological filters, namely climate, disturbance regimes, and biotic interactions [14,15]. Through the metrics of phenotypic characteristics of organisms, it is possible to understand how organisms respond to natural environmental variability or habitat modifications caused by anthropogenic processes [16,17].

Functional homogenization can increase the vulnerability of ecosystems to disturbances and reduce the variability of responses, potentially impacting their resilience and the ecosystem services they provide [18,19]. For example, changes in the composition and activities of soil invertebrates, such as earthworms, beetles, ants, and mites, can influence crucial ecosystem processes like nutrient cycling, organic matter decomposition, and soil structure maintenance [20,21]. Ants are considered suitable bioindicators [22], playing an essential role in ecosystem functional processes, as they assist in nutrient cycling and other crucial ecosystem processes, such as predation, decomposition, and seed dispersal [23,24,25], reacting in various ways to human-induced disturbances, altering their richness and composition [26,27,28].

The use of ant morphology as a proxy for ecological function provides a well-established framework for understanding the effects of anthropogenic disturbances on ant diversity [29,30,31,32,33]. For instance, predatory ants often exhibit small, laterally positioned eyes, a morphological feature that aligns with their hunting behaviors [34,35]. Additionally, the size grain hypothesis, which posits that ants in complex interstitial habitats benefit from relatively short legs for efficient movement, while simpler, planar environments favor longer legs, has been documented in different environments [36,37,38,39]. However, similar morphometric traits may not always indicate similar functions, as one trait can serve different purposes across various ecological contexts [40]. Independently of the trait, the investigation of ant morphological diversity has predominantly focused on local or regional biogeographical levels, with relatively little emphasis placed on the influence of the surrounding environment [29,41,42,43,44].

In this study, we investigated the morphological structure of ant species assemblages in two contrasting environments (agriculture and secondary forests) embedded in two different contexts: a well-established agricultural landscape in Italy and a relatively recent agricultural frontier in the Brazilian Amazon. Contrary to the Brazilian Amazon, where the landscape is still transforming [28], agricultural intensification occurred some decades ago in European ecosystems [41]. Thus, examining the morphological diversity of ants in agricultural areas and secondary forests in Italy can provide insights into the future of the morphological structure of Amazonian ant assemblages under a heavy agriculture scenario. We hypothesize that the effect of environmental filtering on ant assemblages in monoculture areas will be stronger than secondary forests in both countries, leading to greater morphological homogenization and making the assemblages of Italy and Brazil morphologically indistinguishable. However, this effect will be milder in secondary forest areas, whereas the morphological diversity of ant assemblages in Italy will be lower than in Brazil.

## 2. Materials and Methods

The study was conducted in two areas: Ticino River Natural Park in Italy, and the Paragominas mosaic in Pará, Brazil. Both regions encompass forested and agricultural areas, which were the focal points of this work. Ticino River Natural Park (45°35′35.14″ N, 8°43′44.22″ E) is located in northwestern Italy, covering an approximate area of 97 km^2^, serving as an ecological corridor between the mountains of the Apennines and the Alps, crossing the Po River floodplain—one of the most urbanized zones of the country [45]. This large natural area features a mosaic of ecosystems and is recognized as a UNESCO Biosphere Reserve [45]. Paragominas (2°59′51″ S, 47°21′13″ W) is located in the eastern Brazilian Amazon, in Pará. Paragominas was originally covered by perennial tropical forests but has experienced approximately 35% forest loss and widespread degradation of remaining forests in recent decades, primarily due to the conversion of forests into pastures and areas for intensive agriculture. Both sites are immersed in contrasting anthropogenic pressures [28].

The ants sampled in Paragominas were sourced from collections conducted by Solar et al. [28] using baited epigean pitfall traps. The sampling areas encompassed a deforestation gradient, including undisturbed primary and secondary forests that had experienced tree cutting and burning and reforested areas, pastures, and mechanized agricultural sites. This study focused only on two habitats sampled by Solar et al. [7]: secondary forests and areas subjected to mechanized agriculture. Ants were sampled in 15 agricultural and 13 forested sites. A transect comprising six unbaited pitfall traps spaced 50 m apart was established at each location. The traps consisted of 400 mL plastic containers containing a 70% alcohol solution and were left in the field for 48 h. A cover was placed over each pitfall to prevent the entry of leaves and rainwater. Overall, in Paragominas, 90 pitfalls were placed in agricultural areas and 78 in forests.

In Ticino River Natural Park, we sampled ants in 15 agricultural and 12 forested sample site areas. These sites were, on average, 529 m distant from each other (SD: 230, min: 83, max: 1387). Specifically, agricultural sites were, on average, 772 m apart (SD: 404, min: 392, max: 1860), whereas forested sites were 598 m distant (SD: 115, min: 491, max: 937). We followed the same sampling design used by Solar et al. [7]. One transect with six 400 mL unbaited pitfall traps spaced 50 m apart was set up in each location for 90 pitfalls in agricultural areas and 72 in forest areas. Subsequently, the ants were preserved in 90% alcohol in plastic Eppendorf-type containers and identified at the species level with the assistance of a stereomicroscope and based on Radchenko and Elmes [46], Seifert [47], and Csősz et al. [48,49] using specialized papers and reference material in the Entomological Collection of the Instituto Nacional de Pesquisas da Amazônia (INPA), Manaus, Brazil. Vouchers of Italian ant species are deposited in the INPA Entomological Collection. Given the sampling method, we focused only on ground and epigeic species. Pitfall traps provide a reasonable view of the diversity of ground-dwelling ant species. Therefore, arboreal and cryptic litter species are largely missing from our dataset.

Based on the evolutionary life history of ants, we selected seven morphological traits: head length, head width, interocular distance, mandible length, eye width, Weber’s length, and tibia length [50]. These traits cover various functions of ants in the environment (Table 1). We followed the recommendations proposed by the Global Ants project [51] to make the trait measurements. When possible, we used six specimens per species for measurements. Since it is not expected to obtain specimens of all sizes in highly polymorphic genera (i.e., *Pheidole*, *Camponotus*), we measured minor workers among all species. In the region of Paragominas, Pará, Brazil, 1350 worker ants belonging to 264 species were measured. Meanwhile, regarding the ants found in Ticino Park, Italy, 77 ants belonging to 23 species were measured. We used the average measure per trait and species in the further analyses. In total, 18,110 measurements were taken.

The measurements were performed using a Leica 205A stereomicroscope coupled with a Leica DMC4500 camera and the Leica application (version V4. 10. 0) for interactive assembly measurements whenever possible. Measurements were also conducted using the Image J measurement software (version 1.54).

We calculated the Community Weighted Mean (CWM) for each ant trait measured. CWM corresponds to a weighted average of a specific attribute concerning the abundance of all species in each location [52]. This approach provides a more representative measure of the trait under study, considering the relative contribution of each species weighted by relative abundance to each community average trait [52]. Given ants are colonial organisms, we used the occurrence of species at pitfalls as a measure of ant abundance. Therefore, the abundance of a given species varied from zero to six (a given species was sampled in all pitfall traps in a given transect).

To better describe the taxonomical diversity, we constructed species accumulation curves per transect for each habitat and country [53]. Rarefaction or accumulation curves provide good estimations of diversity related to sampling effort. These analyses used the iNEXT package [54], using Hill numbers q = 0, in R (R Core Team 2024. R: A Language and Environment for Statistical Computing_. R Foundation for Statistical Computing, Vienna, Austria. https://www.R-project.org/ (accessed on 6 April 2024)).

We investigated the morphological structure of edaphic ant assemblages at the local scale (transect) and the regional scale (environment). At the transect scale, we compared the CWM of each trait per transect between regions using Hedges’ g effect size. The comparisons were only within habitats (agriculture in Brazil vs. agriculture in Italy and secondary forest in Brazil vs. secondary forest in Italy). This method was chosen because of its ability to provide sensitive and accurate estimates of effect size, considering both the mean difference between groups and within-group variability [55]. Hedges’ g is especially relevant in ecological studies, where small changes can have large impacts and samples often have logistical limitations [56]. Additionally, Hedges’ g is known to correct for slight sample bias [55]. These analyses were calculated using the effect size package [57], in R (R Core Team 2024).

At the regional scale, we estimated the convex hull volume to describe the minimum multidimensional volume encompassing all species within each ant assemblage [58]. We used the seven Principle Coordinate Analysis (PCoA) axes to compute the convex hull of each habitat within the countries. The PCoA was constructed based on the centered (mean of 0 and variance of 1) traits. Given the larger species pool in Brazil (especially in secondary forests), we implemented a bootstrapping procedure to control for the differences in the number of species between countries. Italian and Brazilian agricultural datasets were subsampled to 10 species 999 times without replacement, and Italian and Brazilian secondary forests datasets were subsampled to 14 species 999 times without replacement. The morphological volume was recalculated in each procedure to generate 95% confidence intervals. The convex hulls were calculated using the FD package [59] (H. Wickham. ggplot2: Elegant Graphics for Data Analysis. Springer-Verlag New York, 2016.). All graphs were created with ggplot2 (H. Wickham. ggplot2: Elegant Graphics for Data Analysis. Springer-Verlag New York, 2016.).

## 3. Results

The Paragominas region had 281 species/morphospecies belonging to 52 genera and eight subfamilies (Table 2). The genus with the most species/morphospecies was *Pheidole* (65), followed by *Solenopsis* (23) and *Camponotus* (18). In this region, 81 species and morphospecies of ants were identified in agricultural areas, while in the secondary forest, 166 species and morphospecies of ants were cataloged.

Meanwhile, 23 species of ants were collected and distributed among 15 genera and four subfamilies in the Italian agricultural landscape. *Temnothorax* was the most diverse genera with four species (Table 2). We recorded 13 ant species in agricultural areas in Italy and 17 species in the Italian forest.

Brazil shows a higher species richness than Italy in both habitats (Figure 1). As the number of transects increases, the difference in species richness in agricultural areas between the two countries becomes more pronounced, with Brazil potentially reaching over 30 species. At the same time, Italy’s curve flattens around 15 species for agricultural lands. In the secondary forests, the species richness is significantly higher in both countries compared to agricultural areas. Brazil again shows greater species richness, reaching over 100 species. While more diverse in species richness compared to its agricultural areas, Italy still has a much lower species richness, maxing out at around 60 species.

In general, the CWM of Brazilian ants was higher than that of Italian ants. However, the CWM of agricultural areas was more similar between the two countries. Only the hind tibia length of the Brazilian assemblages was greater than expected by change (Figure 2). Conversely, the CWMs of secondary forests were much more variable, indicating less homogenization between these contexts. Only head width and interocular distance were similar between countries. For the rest of the traits, Brazilian ant assemblages from secondary forests presented a higher CWM than those from Italy (Figure 2).

The morphological space of Brazilian and Italian ants in agricultural areas was similar. On the other hand, we noticed a significant difference in the morphological space of ants between the secondary forests of Brazil and Italy. More specifically, we found that Italy represents a subset of the morphological space of ants found in secondary forests in Brazil (Figure 3). This pattern holds even controlling for the differences in species number between countries (Figure 4). The convex hull of Brazilian secondary forests is still larger than Italian secondary forests when both assemblages have the same number of species.

## 4. Discussion

Habitat disturbances and agricultural intensification can substantially alter community structure and functional diversity [1,2]. As hypothesized, our results demonstrate a convergent morphological homogenization in agricultural landscapes in Italy and Brazil. In both countries, agricultural areas exhibited more morphologically uniform ant assemblages than secondary forests at transect and landscape scales. This pattern is consistent with biotic homogenization, where regional biotas become increasingly similar due to anthropogenic pressures such as habitat conversion and simplification [5,60]. Morphological homogenization, characterized by the reduction in morphological trait diversity, further underscores the ecological consequences of intensive land use [9,10], leading to greater morphological homogenization and potentially compromising ecosystem resilience [19].

Our study underscores the role of environmental filtering in shaping ant communities across contrasting land use scenarios, suggesting that monoculture agricultural systems impose stronger environmental filters on ant assemblages. Overall, Brazilian ants tend to be larger than Italian ants. However, except for hind tibia length, Brazilian and Italian ant assemblages from agriculture were indistinguishable, suggesting a morphological trait convergence likely driven by similar environmental filters from agricultural practices and management. Conversely, secondary forests displayed greater variability in CWM values, reflecting less homogenization among landscapes. These findings underscore the critical role of habitat type in shaping the morphological structure of ant communities. The higher structural complexity and diversity of microhabitats present in secondary forests probably allow the coexistence of species with a broader range of morphological adaptations.

Although there is a similarity in ant space between Brazil and Italy in agricultural areas, our data revealed that the convex hull of ant assemblages in Brazilian secondary forests is consistently larger than that of Italian assemblages, with the latter representing a subset of the morphological space found in Brazil. This pattern holds even when we control for differences in the number of ant species between countries. As expected, Brazilian secondary forests harbor a broader and more complex morphological diversity than their Italian counterparts. This finding can be attributed to the greater environmental heterogeneity and diversity of microhabitats in secondary tropical forests, which foster the coexistence of species with a wide range of functional characteristics. Different types of land use impact arthropod communities in various ways [61], and biomes with higher environmental heterogeneity support greater functional diversity due to the availability of niches [62].

These differences in morphological space highlight the importance of considering local contexts when studying communities from different biogeographical origins [61,62]. While agricultural areas tend to homogenize the morphological characteristics of species due to similar agricultural practices and simplified habitats, secondary forests exhibit greater morphological variation, reflecting the complexity of forest habitats. Some ant species with specific functional traits are favored in areas with intensive land use, such as agricultural and urban areas [63]. On the other hand, due to the rapid transformation of the landscape, some specialized species or functional groups may have their populations drastically reduced or even disappear from the region [64,65,66]. Future studies should investigate the long-term impacts of landscape transformation on the persistence of specialized or generalist species and functional groups, particularly in the context of varying biogeographic regions and land-use practices.

Greater morphological diversity in Brazilian secondary forests may have significant implications for ecological resilience and ecosystem services, as greater morphological variation is often associated with greater stability and responsiveness to disturbances [67]. Although secondary forests may not maintain the same functional diversity as primary forests [29], secondary forests may act as reservoirs of taxonomical and functional diversity in heavily modified landscapes. Furthermore, while secondary forests are increasingly recognized for their ecological contributions, they are often not afforded the same level of protection as primary forests, leading to a disparity in conservation efforts [68,69].

Intensified land use can alter the morphological composition of arthropod communities, affecting services such as organic matter decomposition, nutrient cycling, pollination, and pest control [70]. Our results highlight the importance of secondary forests in agricultural landscapes since these areas maintain a significant morphological diversity of ants that may contribute to essential ecosystem services [71,72,73]. Greater morphological diversity is associated with greater recovery capacity and resistance to environmental disturbances, essential for ecosystem resilience [74,75]. Therefore, conservation strategies that protect secondary forests are crucial to ensuring these ecosystems’ long-term stability and health, especially in the face of increasing anthropogenic pressures and climate change.

## Figures and Tables

**Figure 1 insects-15-00961-f001:**
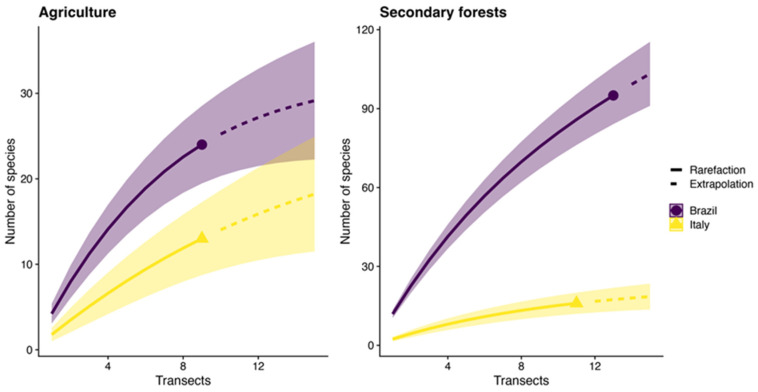
Species rarefaction curves for ants sampled in agricultural and secondary forests in Brazil and Italy. The solid and stippled lines represent the interpolated and extrapolated values, respectively. Shaded areas around the lines represent the 95% confidence intervals.

**Figure 2 insects-15-00961-f002:**
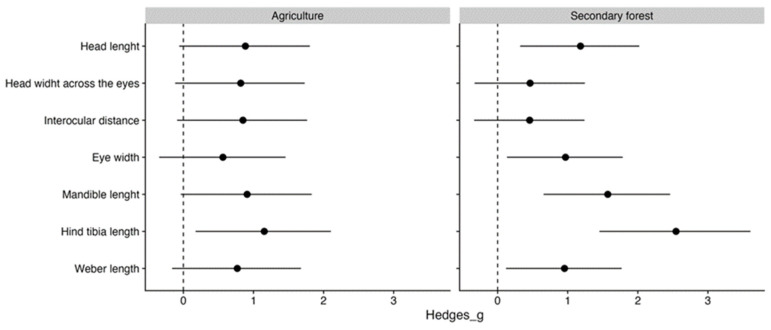
Mean (±95% confidence intervals) effect sizes (Hedge’s g) between Brazilian and Italian CWM for seven morphological traits measured for two distinct contexts. The dashed line represents the Italian CWM. When 95% CI did not overlap 0 (Italian CWMs), the difference in CWM was significant. Points on the right side of the dashed line indicate that the CWM of Brazilian ants is larger than the Italian ant species.

**Figure 3 insects-15-00961-f003:**
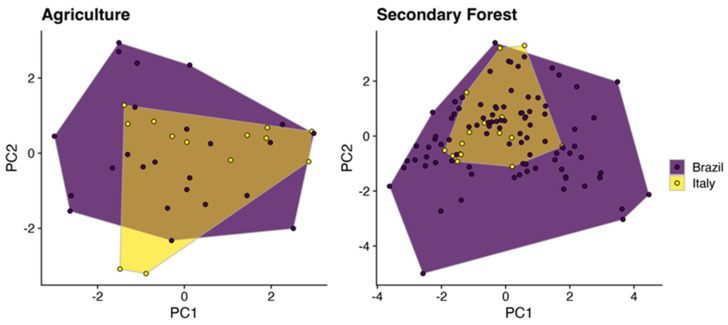
Convex hull projection of morphological trait space for Brazilian (purple) and Italian (yellow) agricultural and secondary forest ant communities.

**Figure 4 insects-15-00961-f004:**
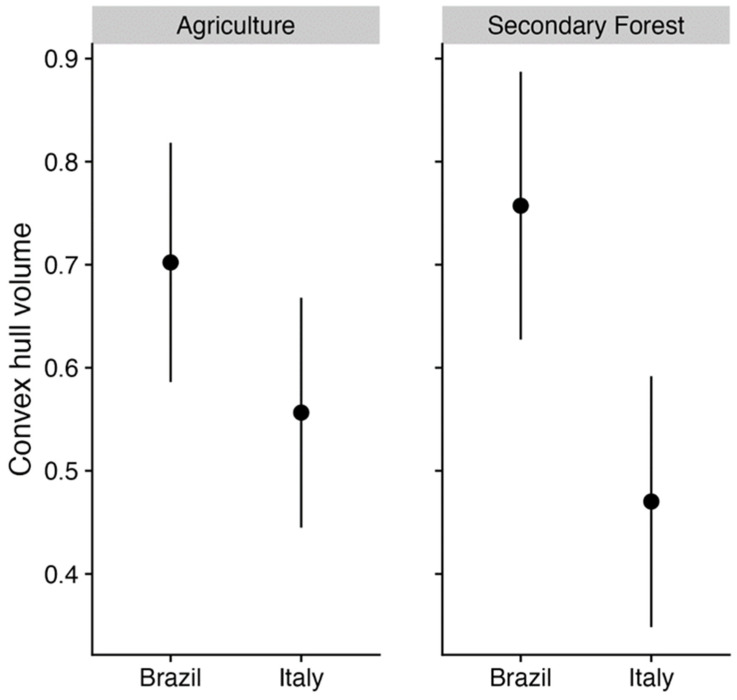
Convex hull volumes of traits between two land uses. The points indicate the means of convex hull volumes. The bars represent 95% confidence intervals calculated after 999 bootstrap resampling.

**Table 1 insects-15-00961-t001:** Morphological characteristics analyzed and their suggested functional significance.

Morphological Traits	Functional Meaning
Head Length Head Width	Head length is an important measurement for understanding the shape and size of a species’ head. It may be related to prey capture, defense, or other ecological functions (Kaspari, 1993).
Interocular Distance	The distance between the eyes is also related to vision and spatial perception. It may reflect the importance of stereoscopic vision in a species (Fowler et al., 1991).
Mandible Length	Mandible length is associated with feeding and capturing prey. Species with long jaws may specialize in capturing larger prey (Weiser and Kaspari, 2006).
Eye Width	Eye width is relevant to vision and perception of the environment. It may indicate the adaptation of a species to different light conditions or the detection of prey or predators (Weiser and Kaspari, 2006).
Weber’s length	Measurement that may be related to body size or the relationship between different parts of the body and associated with resources (Kaspari and Weiser, 1999)
Tibia Length	Tibia length may be related to ants’ functional adaptations, such as feeding behavior, mobility, habitat preferences, and foraging strategies (Gibb and Parr, 2010).

**Table 2 insects-15-00961-t002:** Ground-dwelling-ant species sampled by pitfall trapping in sampling in the Paragominas in the Brazilian Amazon and Ticino River Natural Park in the Italian secondary forest and agricultural areas.

Location	Paragominas		Subfamily/Taxon	Italy	
Subfamily/Taxon	Agriculture	Secondary Forest		Agriculture	Secondary Forest
*Atta cephalotes*		1	*Camponotus lateralis*	1	
*Atta sexdens*	1		*Cardiocondyla elegans*	1	
*Azteca* sp. 3		1	*Crematogaster scutellaris*		1
*Brachymyrmex* sp. 1		1	*Dolichoderus quadripunctatus*		1
*Brachymyrmex* sp. 2	1	1	*Formica cinerea*	1	
*Brachymyrmex* sp. 4		1	*Formica cunicularia*	1	1
*Camponotus atriceps*		1	*Lasius niger*	1	1
*Camponotus blandus*		1	*Lasius myops*	1	1
*Camponotus leydigi*	1		*Monomorium monomorium*	1	1
*Camponotus novogranadensis*		1	*Myrmecina graminicola*	1	1
*Camponotus renggeri*		1	*Myrmica rubra*		1
*Camponotus senex*		1	*Myrmica sabuleti*	1	1
*Camponotus* sp. 15		1	*Myrmica hellenica*		1
*Camponotus* sp. 3		1	*Plagiolepis pygmaea*	1	
*Camponotus* sp. 4		1	*Ponera testacea*		1
*Camponotus* sp. 8		1	*Ponera coarctata*		1
*Carebara brevipilosa*		1	*Solenopsis* cf. *fugax*	1	1
*Carebara escherichi*		1	*Tapinoma subboreale*	1	1
*Carebara lignata*		1	*Temnothorax flavicornis*	1	1
*Carebara urichi*	1		*Temnothorax unifasciatus*		1
*Ceplalotes cordatus*		1	*Temnothorax parvulus*		1
*Crematogaster brasiliensis*		1	*Temnothorax apenninicus*		1
*Crematogaster erecta*		1	*Tetramorium caespitum*-complex	1	1
*Crematogaster flavosensitiva*		1			
*Crematogaster limata*		1			
*Crematogaster sotobosque*		1			
*Crematogaster* sp. 3	1				
*Crematogaster* sp. 5	1	1			
*Cyphomyrmex laevigatus*		1			
*Cyphomyrmex rimosus*		1			
*Dinoponera gigantea*		1			
*Dolichoderus bispinosus*		1			
*Dorymyrmex goeldii*	1				
*Dorymyrmex* sp. 1		1			
*Dorymyrmex* sp. 2	1				
*Dorymyrmex spurius*	1	1			
*Ectatomma tuberculatum*	1	1			
*Gigantiops destructor*		1			
*Gnamptogenys acuminata*	1				
*Gnamptogenys moelleri*		1			
*Gnamptogenys striatula*		1			
*Gnamptogenys tortuolosa*	1				
*Gracillidris pombero*	1	1			
*Hypoponera* sp. 1		1			
*Labidus mars*	1				
*Labidus praedator*		1			
*Labidus spininodis*		1			
*Linepithema neotropicum*		1			
*Mayaponera constricta*		1			
*Mycocepurus smithii*	1	1			
*Neivamyrmex* sp. 2	1				
*Nylanderia* sp. 2		1			
*Nylanderia* sp. 3		1			
*Nylanderia* sp. 4		1			
*Nylanderia* sp. 5		1			
*Nylanderia* sp. 7		1			
*Nylanderia* sp. 8		1			
*Odontomachus brunneus*		1			
*Pachycondyla crassinoda*		1			
*Pachycondyla harpax*		1			
*Pheidole* sp. 01	1	1			
*Pheidole* sp. 02	1				
*Pheidole* sp. 04		1			
*Pheidole* sp. 06		1			
*Pheidole* sp. 08	1	1			
*Pheidole* sp. 11		1			
*Pheidole* sp. 13		1			
*Pheidole* sp. 16		1			
*Pheidole* sp. 17		1			
*Pheidole* sp. 20		1			
*Pheidole* sp. 24		1			
*Pheidole* sp. 27		1			
*Pheidole* sp. 29		1			
*Pheidole* sp. 30		1			
*Pheidole* sp. 31		1			
*Pheidole* sp. 32		1			
*Pheidole* sp. 33	1	1			
*Pheidole* sp. 34		1			
*Pheidole* sp. 35		1			
*Pheidole* sp. 40	1	1			
*Pheidole* sp. 43		1			
*Pheidole* sp. 45		1			
*Pheidole* sp. 49		1			
*Pheidole* sp. 50	1	1			
*Pheidole* sp. 52		1			
*Pheidole* sp. 54		1			
*Pheidole* sp. 57		1			
*Pheidole* sp. 58		1			
*Pheidole* sp. 63		1			
*Pseudomyrmex* sp. 1		1			
*Pseudomyrmex* sp. 3		1			
*Pseudomyrmex termitarius*	1	1			
*Sericomyrmex parvulus*		1			
*Sericomyrmex* sp. 1		1			
*Solenopsis geminata*		1			
*Solenopsis* sp. 1	1				
*Solenopsis* sp. 11		1			
*Solenopsis* sp. 13		1			
*Solenopsis* sp. 16	1	1			
*Solenopsis* sp. 19		1			
*Solenopsis* sp. 2		1			
*Solenopsis* sp. 20		1			
*Solenopsis* sp. 4		1			
*Solenopsis* sp. 6		1			
*Solenopsis* sp. 7		1			
*Solenopsis* sp. 8		1			
*Solenopsis* sp. 9		1			
*Strumygenys denticulata*		1			
*Strumygenys eggersi*	1				
*Strumygenys elongata*		1			
*Strumygenys grytava*		1			
*Strumygenys urrhobia*		1			
*Tapinoma melanocephalum*		1			
*Trachymyrmex bugnioni*		1			

## Data Availability

The datasets generated and analyzed during the current study are available from the corresponding author upon reasonable request.

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
