# Peer review of "How Habitat Simplification Shapes the Morphological Characteristics of Ant Assemblages (Hymenoptera: Formicidae) in Different Biogeographical Contexts"

_insects, 2024, doi:10.3390/insects15120961_

Round 1
Reviewer 1 Report
Comments and Suggestions for Authors
Dear Editor and author,
Thank you for recommending me to review the manuscript. The text is well presented and, given the proposal, the data are well analyzed and discussed. However, the author needs to justify more precisely the reasons that led him to compare such different areas, even though they practice agriculture and have secondary forests. The agricultural areas in Italy are very different from those chosen in Brazil, even in terms of the length of time agriculture has been practiced; and the author writes this in the text. Agricultural management (e.g. agrochemicals, types of cultivation, use or not of agricultural machinery), water regime, soil type, among other characteristics, are different between the countries. The secondary forests in the two countries are completely different. In Brazil, secondary forests have been deforested three times and have suffered fires. However, the text does not provide any such information for the forests of Italy. The list of species confirms the difference between the sites. I recognize that the author has taken the discussion down an interesting but predictable path. In other words, there is little new in the discussion. However, the author's finding that Italy represents a subset of the morphological space of ants found in secondary forests in Brazil (line 224) is an unexpected result.
Here are some comments:
1. keywords - give preference to using words that are not in the title or summary.
2. Line 29: the information is for Brazil only. See lines 122 and 128.
3. Lines 40 to 41: what is the research question?
4. Line 84 - Silva et al. (2022) did it along the Atlantic Forest. I suggest reinforcing it, as it was neither a local nor a regional study.
5. Lines 85 to 87 - I didn't understand the information regarding the objective and hypotheses of the work.
6. Lines 91 to 92: the author made it clear that the agricultural areas are very different from each other. But what is the research question?
7. Lines 96 to 97: Even though agricultural areas in Brazil are more recent in the place studied, the same impacts should be observed. So why did you only emphasize the areas in Italy? In addition, your work does not propose to relate climate change and the loss of ecosystem services. So I don't understand why this information was included here.
8. Line 98: put a period after [41]
9. Lines 117 - 120 - I don't find the information necessary. It's enough to say where the samples were taken.
10. Lines 124 and 130: mL
11. Lines 133 and 134: what literature was used to identify the species in Brazil? Were the vouchers from Brazil and Italy deposited at INPA?
12. Lines 138 to 140; 152 to 154: I don't understand.
13. Lines 187 and 192: put the information in taxonomic sequence.
14. Lines 188, 189, 193: genus in italics
15. Line 196: do not include discussion in the results.
16. Line 206: standardize rarefation and accumulation - caption in graph and figure title.
17. Line 238 to 246 - indicate whether the hypotheses were achieved or not.
18. Line 248 to 253 - check
19. Lines 285 to 287 - I didn't understand the context.
20. Lines 291 to 300 - review
21. Lines 340, 467: review abbreviation
22. Table 1 and 2: supplementary material
23. Figure 3 - explore further in the discussion
24. References: standardize (e.g. journal abbreviations)
Reviewer 2 Report
Comments and Suggestions for Authors
The study investigates how patterns of biotic homogenization vary across continental scales and among habitat types, specifically as a result of anthropogenic pressures, such as land use. The study is elegantly conducted and highlights an important effect of intensive land use which is consistent across continents. I have added a few minor suggestions for improvement. I would recommend paying attention to the abstract. read it through the eyes of someone who hasn't read the article and isn't familiar with the methods.
in the attached Word file are some minor comments.

There are a few minor sentences where the wording can be improved. This has been addressed in the attached.
Reviewer 3 Report
Comments and Suggestions for Authors
This paper deals with a comparison between ant samples from two regions (Italy vs. Brazil), in either region comparing samples from agricultural land with secondary forests. The basic idea is to use measurement data on 7 morphometric traits to explore whether ant assemblages differ systematically along the two contrasts (biogeographical vs. regional land-use). One might ask whether it is really warranted to compare these two regions (the choice is completely arbitrary and not guided by a convincing scientific rationale), but as a case study I am inclined to accept that choice.
As expected, there are substantial differences in ant morphology between Italy and Brazil, which is not surprising given the far higher phylogenetic diversity of ants in the tropics (far more subfamilies, tribes and genera being represented in the tropics). In either region, the occupied morpho-space tended to be contracted in ant assemblages on agricultural land as opposed to secondary forest.
However, I am truly sceptical about equating these morphometric results with ‘functional diversity’, as is done in the title and throughout the whole paper. As is correctly stated in the paper (e.g. in the heading of the supplementary data) these selected traits just have a ‘suggested’ (i.e. not ‘proven’) functional significance, which might also not be consistent across ant taxa. For example, long mandibles could be related to very specific predatory behaviour, but also (as in Odontomachus) in a kind of jumping mechanism: V Mohan & JC Spagna (2015). Jump performance in trap-jaw ants: beyond trigger hairs. Bull. NJ Acad. Sci, 60(2), 1-4.), or be useful in stealing slave pupae (like in the famous genus Polyergus). In other words: the same or similar expression of one morphometric trait does not necessarily indicate similarity in function. From the perspective of the functional role ants may play in ecosystems, other traits are for sure far more important. For example, whether leaf-cutters, army ants, fungus growers, mound-building red wood ants, or seed harvesters are more prevalent in forest or in agricultural land would provide far deeper insight into functional impoverishment (or functional homogenization). Similarly, for the functional diversity among ant assemblages, traits like colony size, position in the continuum herbivore/omnivore/predator, extent of intraspecific worker polymorphism etc. are for sure more significant than are subtle variations in some body metrics.
I therefore urge to use a more honest wording, e.g. “occupancy of morphological trait space” (this is what the authors have truly studied!) rather than “functional diversity”.
The title of the paper is also misleading in another way, as it states something about “changes in land use”. However, I could not find any results in the paper which would address such ‘changes’ in space or time, for example different intensities of agriculture or forestry. Rather, they compare two (insufficiently defined, see below!) classes of land use. Again, this can easily be fixed by adopting an appropriate terminology that adequately describes what has truly been done. This criticism does NOT de-value the present work, but I seriously question how this work is being ‘framed’. My impression is that the authors ‘over-sell’ their (nice) results.
The study sites where ants have been sampled must also be much better described and defined. For example, how old were secondary forests? How much time has passed since their last use for forestry purposes or since they have been planted or left to spontaneous succession? How tall are the trees, how many vegetation layers can be recognized, what about canopy closure, etc.? For sure, a 5 years old secondary forest offers a different kind of habitat to ants than a 50 years old one, but still they are both ‘secondary forests’. Obviously, to make comparisons between Italy and Brazil meaningful, forest sites should be selected in such a way that they are structurally comparable. The same applies to agricultural land. In the absence of further specification, this could comprise anything from annual crop fields (soy beans, potatoes, cereals or maize) to perennial crops (grapes, bananas, fruit trees, coffee) with dramatically different opportunities for ants. So please describe the selected agricultural habitats in more detail. Also, a map or aerial photograph must be given in the supplementary material where the locations of the sites within each region can be seen by the reader. Otherwise one cannot get an idea about the representativeness of the ant samples.
With regard to the ant species list, I wonder how sure you can be about the large numbers of un-identified species in genera such as Pheidole, Camponotus, Crematogaster, Nylanderia, or Solenopsis. Did you check species differences by DNA barcoding, or by multivariate morphological scrutiny (e.g. discriminance analyses)? I agree that in these genera, species identifications may still be impossible (for example, because of undescribed species). But given the very large contribution of these OTUs to the species list, the process by which ant species were sorted needs to be more carefully documented (in the supplementary materials), because this could have a major impact on the results. Did you use Wilson’s Pheidole monograph? (EO Wilson (2003). Pheidole in the New World: a dominant, hyperdiverse ant genus Harvard University Press.). Or this one for Nylanderia: JS LaPolla et al. (2011). Monograph of Nylanderia (Hymenoptera: Formicidae) of the world: an introduction to the systematics and biology of the genus. Zootaxa, 3110(1), 1-9. I noted a few mis-spellings of ant taxon names (e.g. Strumygenys should read Strumigenys; Ceplalotes should read Cephalotes).
Overall, the paper will benefit from a careful proof reading for English grammar and style, I noticed quite a few relevant places.
Below, I list a number of further specific points that must be addressed during revision of the paper.
L 39: which aspects of ant morphology? Be concrete in the abstract!
L 45: CWMs of what type of trait(s)? Must be explicitly said.
L 47: convex hull of what? No reader can understand that.
L 67: should read: ‘complementary’
L 77: indicators for what? Please say specifically what can be indicated using ants, and why this might be preferable to using other indicators or metrics. I frequently get the impression, as a reviewer, that some scientists just use "bioindicator" as a "buzzword" to make their papers appear “more interesting”. Honestly, I do not require ants as (taxonomically difficult) “indicators” to tell apart a secondary forest from agricultural land.
L 88: there are quite a number of papers addressing such questions in trans-regional or other large-scale studies, besides what you have cited e.g.
https://doi.org/10.1002/ece3.10000
https://doi.org/10.1371/journal.pone.0064005
DOI: 10.1111/1365-2435.14135
L 118: perhaps say in one sentence that no litter samples were secured, nor were arboreal ant species addressed. Also nest inquiline social parasites will not be covered by samples from both regions. Nothing is 'wrong' with that approach, but this means that a certain fraction of functional trait space occupied by ant assemblages was not covered in either region. I also noted that in Brazil you employed baited pitfall traps, yet un-baited ones in Italy? Why? This is a potentially major flaw in your study design, since baited traps may attract substantially more ants. For a comprehensive characterization of the ant assemblages of a region, using just one collection method is usually insufficient: D Agosti et al. (2000). Ants. Standard methods for measuring and monitoring biodiversity. Smithsonian Institution, Washington DC.
L 138-140: delete!! If applicable, please make a statement about collecting permits for your field work in Italy (I don't know whether this is necessary for sampling ants??). I presume the necessary information for Brazil has already been included in Solar et al., but perhaps mdpi publisher’s requirement is to re-iterate that here.
L 146: How did you deal with species that show pronounced worker polymorphism, such as many Pheidole, Camponotus, Atta etc. species? Did you deliberately cover the entire gradient of worker size, or did you focus on minor workers?
L 147: specify the number of measurements: … at altogether XXX worker ants representing YYY species ... Please report (approximate) number of measured specimens and species.
L 152-154: repetitive, delete...
L 160: please be more precise here. I presume, but I may have misunderstood your wording, that you used the incidence of ant species in replicate traps per site, rather than the collected worker numbers, as a proxy for abundance. Under "frequency" one might also understand number of captured worker individuals.
L 166 ff: what software was used for calculating Hedge's g? Please correctly cite all such sources!
L 178: please cite the software used to do the PCoA, the bootstrapping procedure, and the construction of convex hulls; otherwise your study could not be reproduced. Were traits centred to a common scale (mean of 0 and and variance of 1) before entering them into the PCoA?
L 188: please format all genus and species names in italics
L 315/316: usually mdpi Publishers require all raw data to be made publicly available (open science policy), but this is of course at the editor’s discretion.
Comments on the Quality of English Language
Overall OK, but at various places some linguistic polishing would be adisable.
